# A framework for effective face-mask contact modeling based on finite element analysis for custom design of a facial mask

Yun-Jae Kwon[1], Jin-Gyun Kim[1]*, Wonsup Lee[2]*

1 Department of Mechanical Engineering, Kyung Hee University, Giheung-gu, Yongin-si, Gyeonggi-do, Republic of Korea, 2 School of Global Entrepreneurship and Information Communication Technology, Handong Global University, Buk-gu, Pohang-si, Gyeongsangbuk-do, Republic of Korea

* jingyun.kim@khu.ac.kr (JGK); w.lee@handong.edu (WL)

## Abstract

A novel contact model is presented to efficiently solve a face-mask contact problem by using the finite element (FE) method for the optimized design of a custom facial mask. Simulation of contact pressure for various mask designs considering material properties of the face allows virtual evaluation of the suitability of a mask design for a person's face without conducting empirical measurement of the face-mask contact pressure. The proposed contact model is accomplished by combining three approaches to reduce the calculation cost of simulating the face-mask contact: (1) use of a simplified and modifiable mask model that applies a spline curve to design points; (2) reduction of the FE model of the face by applying static condensation; and (3) application of a contact assumption that uses the Lagrange multiplier method. A numerical case study of a medical mask design showed that the proposed model could calculate the face-mask contact pressure efficiently (0.0448 sec per design). In a pilot usability experiment, the measured contact pressure was found similar values (range of mean contact pressure: 0.0093 ~ 0.0150 MPa) to the estimated values (range of mean contact pressure: 0.0097 ~ 0.0116 MPa).

## 1. Introduction

A facial mask must fit properly to sufficiently protect its wearer, so it must be designed considering the characteristics of its contact with the face (face-mask contact). An improperly designed mask can allow air leakage or exert excessive pressure on the face that can lead to discomfort, rashes, and injuries [1–5]. Lee et al. [6], and Schereinemakers et al. [4] emphasized that an oxygen mask for fighter pilots needs a proper fit to the face for safety as the oxygen mask supports an effective and comfortable oxygen supply to the pilot at a high altitude where oxygen is lacking. During the course of the COVID-19 pandemic, many studies on the design and testing of facial masks reported that the fit of the mask to the face has a significant effect on health and safety [7–9]. While the facial mask is commonly used in the daily lives of people, it needs to be properly designed to both protect people from respiratory disease and to provide comfort during long-term use.

Data Availability Statement: The data relevant to this study are available from Figshare at: https://doi.org/10.6084/m9.figshare.20055167.v1 (10.6084/m9.figshare.20055167)."

**Funding:** Acknowledgement - This research was jointly supported by the National Research Foundation of Korea (NRF) grants funded by Korea Government (2020R1F1A1050076 and 2021R1A2C4087079). Initials and grant numbers - WL: 2020R1F1A1050076 - JGK: 2021R1A2C4087079 Full name of each funder - the National Research Foundation of Korea (NRF) (URL: https://www.nrf.re.kr/eng/index) Did the sponsors or funders play any role in the study design, data collection and analysis, decision to publish, or preparation of the manuscript? - NO. The funders had no role in study design, data collection and analysis, decision to publish, or preparation of the manuscript.

**Competing interests:** The authors have declared that no competing interests exist.

A facial mask designed considering anthropometric information and 3D morphological images could increase user satisfaction with fit, comfort, safety, and usability. A 3D scan image of the face has been usefully applied in the ergonomic design of facial masks as it provides various information, including anthropometric facial dimensions (e.g., length, width, circumference, and any other point-to-point or surface lengths) and morphological information (e.g., arc, cross-sectional curvature, surface, area, and volume). Lee et al. [3] developed an ergonomic design of the oxygen mask for Korea Air Force (KAF) pilots based on an analysis of anthropometric characteristics, contact between 3D face scan images (n = 336) and morphological features of the mask. The proposed design showed 33% to 56% lower discomfort, 11% to 33% less contact pressure, and a more satisfying fit for KAF pilots than those of the existing pilot oxygen masks created considering the facial characteristics of U.S. Air Force personnel. Morrison et al. [10] proposed a method of designing a custom mask based on 3D facial scanning and 3D printing techniques to supply oxygen stably to a child who had craniofacial anomalies.

The finite element (FE) analysis method has been applied to consider the deformation of the face and mask for realistic estimation of the face-mask contact characteristics. FE analysis is a computing technique for numerical approximation regarding morpho structural changes of the FE model, which is generated based on a 3D object or a 3D scanned human body part. The FE model is formed based on a finite number of small elements (say, triangulated meshes) and information on material properties (e.g., Young's modulus, Poisson's ratio) [11–13]; and the FE analysis can estimate the biomechanical responses (e.g., deformation, stress, contact pressure) to a particular force applied to the human body [14, 15]. As for the face-mask studies, Bitterman [16] simulated the contact pressure between FE models of a pilot's face and an oxygen mask design, which were prepared through 3D scanning and FE modeling processes. However, the face is simply assumed as a rigid body, unrealistically, while the oxygen mask was modeled as a deformable object in the study. Yang et al. [17] used LS-DYNA, a popular commercial software for FE analysis, to estimate the contact pressure of a respirator worn over a human headform, which was 3D scanned and FE modeled. The respirator is FE modeled as a single-layered shell element (thickness = 2 mm) with silicone-like material properties (Elastic Modulus: 2 MPa, Poisson's Ratio: 0.499, Density: 1 g/cm$^3$); and the FE model of a headform was simplified as a single-layered shell element (thickness = 5 mm) with material properties of the soft-tissue (Elastic Modulus: 2 GPa, Poisson's Ratio: 0.35). Dai et al. [18] and Lei et al. [19] extended Yang et al.'s work by applying a more realistic headform created with a consideration of the multi-layered anatomical structure such as skin, soft tissue (e.g., muscle, fat), and bone. The computational results of the contact pressure were compared with experimental results measured using the Tactilus freeform sensor system (Sensor Products Inc., Madison, N.J.) and found similar results in the contact pressure (mean difference: 0.013 MPa). Lastly, Gök [20] and Gök et al. [21] performed the face-mask contact analysis with a simple FE model, and the results showed the edge shape of a mask can reduce the contact pressure and protect people from respiratory droplets that can cause the infection of the COVID-19.

Despite the reliable results of the face-mask FE analysis, the application in the industries has not been widespread and has a lot of obstacles. As the primary reason, it is difficult to employ the conventional FE analysis in the design process of a consumer product as it costs time and effort to create multi-layered FE models for hundreds or thousands of head data to consider the morphological variabilities of a target user population [3, 27]. Therefore, to increase the utility of this method in the product design process, an accurate and low-cost FE method is required for the practical application of contact estimation to find the best design candidates within a manageable time.

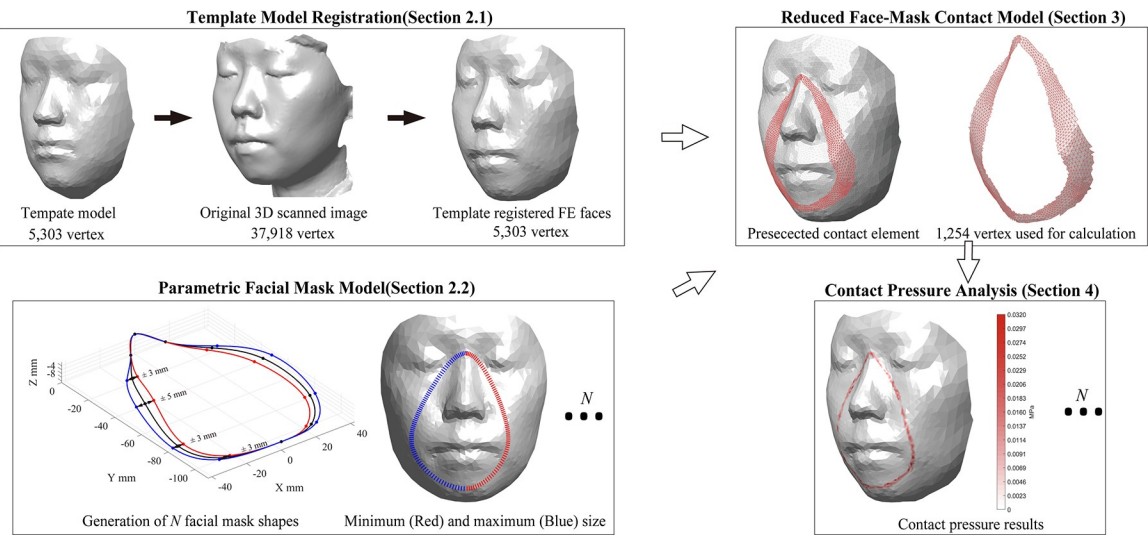

**Fig 1. Concept of a novel design approach for a facial mask based on finite element analysis, illustrated with an example face image.**

This study aimed to develop an effective face-mask contact model which can satisfy the procedural simplicity, computational efficiency, and industrial applicability to design a facial mask based on 3D scanned images of the face and the FE analysis. In the face-mask contact analysis, this study uses the contact pressures over the face calculated according to the shape of the mask, as an indicator of discomfort, air leaking, and usability that are useful information in the product design process. Section 2 (Data preparation for the finite element analysis) introduces FE modeling of the face with simplified anatomical structure and mesh topology and explains the use of a spline curve to simplify the facial mask. Section 3 (Face-mask contact model) presents the mathematical model of face-mask contact analysis, considering the model reduction [22, 23] to accelerate the computational efficiency. Section 4 (Numerical studies) provides a numerical example of the face mask design for one person through a pilot study, and the reliability of the proposed method has been investigated by measuring contact pressure using force-sensitive resistor (FSR) sensors. The essential idea of this study is illustrated in Fig 1.

## 2. Data preparation for the finite element analysis

The FE-based models of the face and a mask are required for further face-mask contact analysis and to search for the optimal mask design. The FE-based face and mask models both consist of 3D morphological (e.g., nodes, elements) and mathematical (e.g., stiffness matrix) features, which are used in the development of the face-mask contact model in Section 3. Techniques of a template model registration (TMR) and generation of a surface from a point cloud are applied to simplify the computation of contact between face and mask. This section presents the theory and process of FE modeling.

### 2.1. FE model of the face

**2.1.1. Preparation of 3D template-registered images of the face.** 3D scanning is used to collect images of the face, and then it is represented as a point cloud of vertices and a triangular mesh that represents connections among them. This process represents a complex geometric object as a set of triangles, and 3D scan images require the editing procedure to repair its mesh

errors (e.g., holes, improper triangle meshes, floating parts, intersecting triangles), in general, using 3D image processing software [24, 25]. Then, 3D locations of anthropometric landmarks (e.g., sellion, the nasal root point at the sagittal plane; ectocanthion, the lateral corner of the left and right eyes; and promentale, the most protruded point of the chin at the sagittal plane) are found on the surface of the 3D face images [6, 25]. Lastly, all 3D face images are aligned based on the anthropometric landmarks; thus, their orientations become the same. Once the 3D face images are post-processed, TMR is applied to the post-processed 3D face images to build a computationally efficient FE model. During the 3D scanning procedure, a fine density of vertex points is randomly generated to form the surface of a target object; therefore, 3D face scan images are inconsistent in the number of vertices and in the topology of the mesh structure [26]. To compare the results of face-mask contact analyses for different faces, the mesh structure of 3D face images needs a reformation to become topologically corresponded to each other by applying TMR [25–27]. Also, TMR can lead to efficient computation in further face-mask contact analysis as the number of meshes as well as the mesh topology of the 3D face images can be optimized. A landmark-based TMR approach proposed by Lee et al. [28] was applied in this study. First, a template face image is roughly aligned to each post-processed 3D face image by adjusting its size based on the anthropometric landmarks using the bounded biharmonic weights (BBW), a mesh deformation algorithm [29]. Then, a non-rigid iterative closest point (ICP), a 3D image registration technique [30], is used to reduce the surficial differences between the size-adjusted template image and each target face scan image. This process yields template-registered face images that consist of a reasonable number of vertices with a particular mesh topology.

**2.1.2. Simplification of the FE model of the face.** In this work, the shell-type elements are used to describe the template-registered face image, which is a thin surface (Fig 2). This approach creates a great advantage in the computational cost as a 3D face image can directly be applied to the FE model without considering anatomical layers such as soft tissue and bone.

Face-mask contact is assumed as a linear elastic and static problem for cost-efficient computation. The FE model is then considered to calculate the static deformation of the face. This task requires a stiffness matrix that represents the geometric data and material properties. The formulation of the FE stiffness matrix can be derived from the following principle of virtual

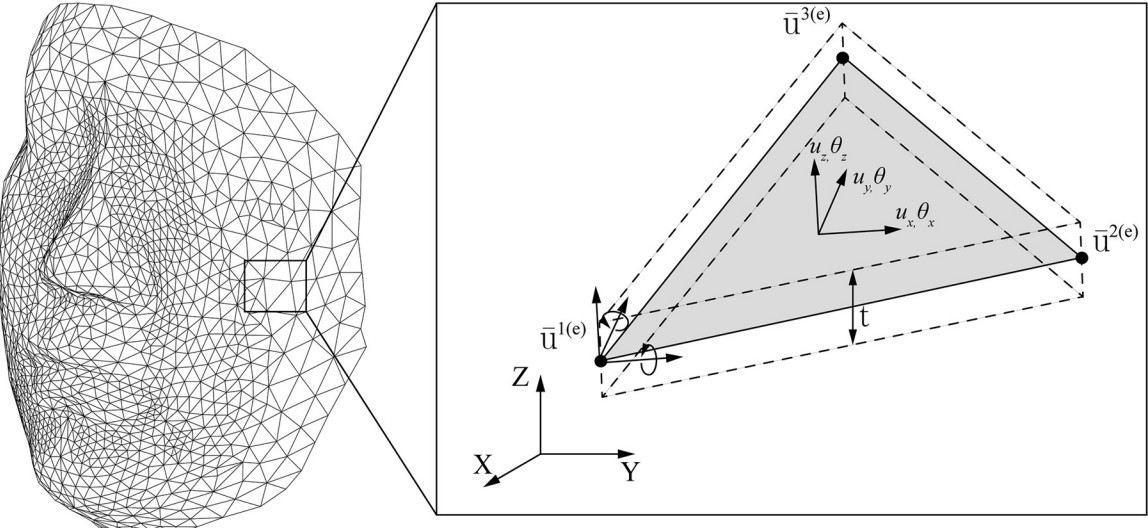

**Fig 2. The template-registered face image concept of a single triangular shell element.**

work:

$$\int_V \delta\epsilon\boldsymbol{\tau}dV = \int_{S_t} \delta\mathbf{v}\mathbf{t}dS_t + \sum_i^{N_i} \delta\mathbf{v}\mathbf{f}_c^{(i)},\tag{1}$$

where $\tau$, $\delta\epsilon$, and $\delta\mathbf{v}$ are stress, virtual strain, and virtual displacement, respectively. $V$ and $S_t$ are body volume and a surface under pressure. $\mathbf{f}$ and $\mathbf{t}$ are the nodal load and surface traction, respectively. Those can describe the external contact force from mask to face in this problem. The $N_i$ is the number of nodal loads. Eq (1) represents the energy equilibrium between internal virtual work (left-had side), and external virtual work (right-hand).

A displacement within an element can be assumed by a polynomial and nodal displacement. The element-wise displacement is defined as

$$\mathbf{u}^{(e)} = \mathbf{H}^{(e)}\bar{\mathbf{u}}^{(e)}, \epsilon^{(\mathbf{e})} = \mathbf{B}^{(e)}\bar{\mathbf{u}}^{(e)},\tag{2}$$

$$\delta\mathbf{u}^{(e)} = \mathbf{H}^{(e)}\delta\bar{\mathbf{u}}^{(e)}, \delta\epsilon^{(e)} = \mathbf{B}^{(e)}\delta\bar{\mathbf{u}}^{(e)},\tag{3}$$

where $\mathbf{H}$ is the displacement interpolation matrix and $\mathbf{B}$ is the strain-displacement matrix. The superscript $e$ denotes a single element number. The shape function can be changed to a selection of the type of FE element.

As presented in Fig 2B, the displacement vector $\bar{\mathbf{u}}$ is a collection of a triangular shell nodal displacement vector and it can be defined as

$$\bar{\mathbf{u}}^{i(e)} = [\bar{\mathbf{u}}_x^{i(e)}, \bar{\mathbf{u}}_y^{i(e)}, \bar{\mathbf{u}}_z^{i(e)}, \bar{\theta}_x^{i(e)}, \bar{\theta}_y^{i(e)}]^\mathrm{T}, i = 1, 2, 3,\tag{4}$$

where $i$ denotes a node number (1, 2, or 3) in a triangle shell element. The displacement of a node can be defined by three transformations and two rotations except the in-plane rotation, and thus the three-node FE shell element has total of 15 degrees of freedoms (DOFs). The DOF implies the number of independent displacements that can express a configuration or a state of a body.

By substituting Eq (2) and (3) into Eq (1), the element-wise virtual work is obtained as

$$\sum_e^{N_e} \int_{V^{(e)}} \mathbf{B}^{(e)T}\mathbf{D}^{(e)}\mathbf{B}^{(e)}dV^{(e)} = \sum_e^{N_e} \int_{S_t^{(e)}} \mathbf{H}^{(e)T}\mathbf{t}^{(e)}dS_t^{(e)} + \sum_i^{N_i} \mathbf{H}^{(e)}\mathbf{f}_c^{(i)},\tag{5}$$

where $N_e$ is the number of elements used to the entire surface of the face. Eq (5) accounts that the response of a body $\bar{\mathbf{u}}$ can be defined as the assemblage of the response of elements, and based on it, the body can be discretized by a set of elements.

The stiffness is defined by the left-hand side of Eq (5). As a simple approach, the shell element can be built by summing plate bending stiffness $\mathbf{K}_b$, plane stiffness $\mathbf{K}_p$, and shear stiffness $\mathbf{K}_s$ as

$$\mathbf{K}^{(e)} = \mathbf{K}_b^{(e)} + \mathbf{K}_p^{(e)} + \mathbf{K}_s^{(e)},\tag{6}$$

$$\mathbf{K}_b^{(e)} = \int_{V^{(e)}} \mathbf{B}_b^{(e)T}\mathbf{D}_b^{(e)}\mathbf{B}_b^{(e)}dV^{(e)},$$

$$\mathbf{K}_p^{(e)} = \int_{V^{(e)}} \mathbf{B}_p^{(e)T}\mathbf{D}_p^{(e)}\mathbf{B}_p^{(e)}dV^{(e)},\tag{7}$$

$$\mathbf{K}_s^{(e)} = \int_{V^{(e)}} \mathbf{B}_s^{(e)T}\mathbf{D}_s^{(e)}\mathbf{B}_s^{(e)}dV^{(e)},$$

where $\mathbf{D}$ is the elasticity matrix for plate bending ($\mathbf{D}_b$), plane ($\mathbf{D}_p$), and shear ($\mathbf{D}_s$) defined as

$$\mathbf{D}_b = \frac{Eh^3}{12(1-v^2)} \begin{bmatrix} 1 & v & 0 \\ v & 1 & 0 \\ 0 & 0 & (1-v)/2 \end{bmatrix}, \mathbf{D}_p = \frac{E}{(1-v^2)} \begin{bmatrix} 1 & v & 0 \\ v & 1 & 0 \\ 0 & 0 & (1-v)/2 \end{bmatrix}, \mathbf{D}_s = \kappa G2, G = E/(2(1+v)), \quad (8)$$

where $v$ is Poisson's ratio, $E$ is the elastic modulus, and $h$ is the thickness of the shell element.

Then, through the assembly process, the linear static equation of the face model can be expressed by

$$\mathbf{K}_f \bar{\mathbf{u}}_f = \mathbf{f}_f, \quad (9)$$

where $\mathbf{K}_f$ is the stiffness of the face, $\bar{\mathbf{u}}_f$ is the displacement of the face, $\mathbf{f}_f$ is the external force applied on the face, and the subscript $f$ denotes the face. According to Eq (9), the deformation of the face FE model $\bar{\mathbf{u}}_f$ is calculated as a solution of FE analysis when the force $\mathbf{f}_f$ is imposed by contact with the mask over the face.

The derivation shows that the configuration of mesh, material properties, boundary condition, and external load are the main parameters of the FE model, and thus the realistic results of the FE model depend on the reliability of these parameters. The proper mesh configuration, one of the important factors affecting accuracy, has been generated by TMR in this study. A Young's modulus of 0.03 MPa and a Poisson's ratio of 0.49 are used as material properties of soft tissue in this study by referring to previous studies [27, 31–37]. The thickness of the shell element was defined as 2 mm. The fixed boundary condition is applied at an open edge of the face (e.g., the boundary of the 3D scanned image of the face), and only an external force is considered as the contact force (Section 3.2).

## 2.2. Parametric model of the facial mask

This study proposes a simply adjustable model of the mask composed of a set of nodes generated along a reference line created based on reference points. The proposed mask model uses three strategies. First, to simplify and reduce the cost of analysis, only the part of the mask that contacts the face is considered. Second, a reference line is created by fitting a spline curve to a few reference points (Fig 3A) to adjust the form of the mask efficiently; therefore, different forms of the mask can be automatically generated and evaluated in a face-mask contact simulation. Lastly, the 3D form of the mask consisting of equally distributed nodes is generated along the reference line. A set of cross-section nodes (Fig 3A) is used to define a custom form of the mask in this study. By placing the multiple cross-sections on normal planes of the reference line with an interval of about 1.0 mm, the 3D form of the mask can be created, as shown in Fig 3B. Note that the number of cross-sections varies according to the size and form of the reference line. The proposed approach of generating the mask's form is used in the FE analysis of the contact between a face and multiple mask designs described in Sections 3 and 4.

A piecewise cubic spline curve [38] was used to express the reference line as a proper interpolation method in this study. A spline curve is a collection of the 3rd-order polynomials; it can represent a complex curve without overfitting. A piecewise cubic spline curve is created at each division, which is defined as an interspace between two reference points of the mask. Note that the number of spline curves is the same as the number of divisions. Eq (10) to (18) sequentially explain the piecewise cubic spline function for the generation of a complete form of the mask.

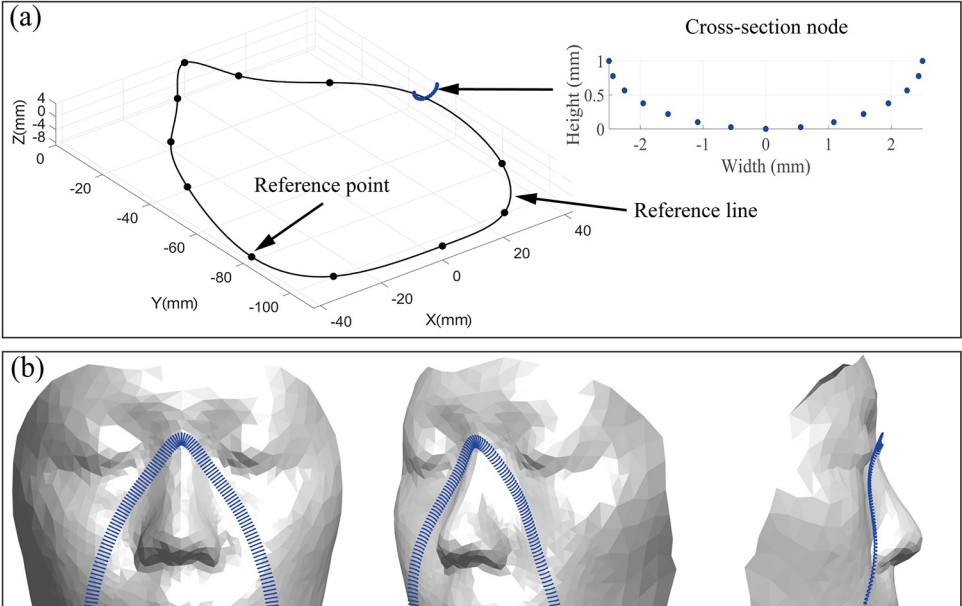

**Fig 3.** (a) Reference line generated considering reference points and an example of cross-section, (b) 3D form of contact surface of a mask, composed of multiple cross-sections.

First, the one-dimensional cubic spline function can be written as:

$$S(p)^{(k)} = a_3^{(k)}p^3 + a_2^{(k)}p^2 + a_1^{(k)}p + a_0^{(k)}, k = 1, 2, \cdots, n-1, \tag{10}$$

where $k$ is the $k$th division between two reference points $p^{(k)}$ and $p^{(k+1)}$, and $p$ is a point in the $k$th division defined as $p^{(k)} \leq p \leq p^{(k+1)}$. $S(p)^{(k)}$ is the $k$th piecewise cubic spline function, which consists of $n-1$ internal points ($k = 1,2,\ldots,n-1$) and two endpoints ($k = 0,n$). Thus, the number of the cubic spline function is $n$. The second derivative of Eq (10) can be expressed as a first-order Lagrange polynomial

$$S''(p)^{(k)} = g''^{(k+1)}\frac{p - p^{(k)}}{l^{(k)}} - g''^{(k)}\frac{p - p^{(k+1)}}{l^{(k)}}, l^{(k)} = p^{(k+1)} - p^{(k)}, \tag{11}$$

where $l^{(k)}$ is the $k$th section length. $g''^{(k)}$ and $g''^{(k+1)}$ are the second-derivative terms at $p^{(k)}$ and $p^{(k+1)}$, respectively. The integration result of Eq (11) can be written as

$$S'(p)^{(k)} = g''^{(k+1)}\frac{(p - p^{(k)})^2}{2l^{(k)}} - g''^{(k)}\frac{(p - p^{(k+1)})^2}{2l^{(k)}} + c_1, \tag{12}$$

$$\begin{aligned} S(p)^{(k)} &= g''^{(k+1)}\frac{(p - p^{(k)})^3}{6l^{(k)}} - g''^{(k)}\frac{(p - p^{(k+1)})^3}{6l^{(k)}} + c_1 p + c_0 \\ &= g''^{(k+1)}\frac{(p - p^{(k)})^3}{6l^{(k)}} - g''^{(k)}\frac{(p - p^{(k+1)})^3}{6l^{(k)}} + A\frac{p - p^{(k)}}{l^{(k)}} - B\frac{p - p^{(k+1)}}{l^{(k)}}, \end{aligned} \tag{13}$$

where $A$ and $B$ are the unknown constants. The two endpoints $p^{(k)}$ and $p^{(k+1)}$ can determine $A$

and *B*. Substituting them into Eq (13) yields Eq (14), which can provide a cubic spline curve.

$$S(p)^{(k)} = g^{''(k+1)}\left[\frac{(p-p^{(k)})^3}{6l^{(k)}} - \frac{l^{(k)}(p-p^{(k)})}{6}\right] + g^{''(k)}\left[-\frac{(p-p^{(k+1)})^3}{6l^{(k)}} + \frac{l^{(k)}(p-p^{(k+1)})}{6}\right]$$
$$+g^{(k+1)}\frac{p-p^{(k)}}{l^{(k)}} - g^{(k)}\frac{p-p^{(k+1)}}{l^{(k)}}. \tag{14}$$

where $g^{(k)}$ and $g^{(k+1)}$ are defined from the reference points, but second derivative terms $g^{''(k)}$ and $g^{''(k+1)}$ are unknown in general. To define them, a continuity condition of all the joint points is considered. The first derivative of cubic splines then fulfills the tangential continuity

$$S'(p^{(k)})^{(k)} = S'(p^{(k)})^{(k-1)}, \tag{15}$$

to yield

$$g^{''(k+1)}l^{(k)} + 2g^{''(k)}[l^{(k)} - l^{(k-1)}] + g^{''(k-1)}l^{(k-1)}$$
$$= 6\left[\frac{g^{(k+1)} - g^{(k)}}{l^{(k)}} - \frac{g^{(k)} - g^{(k-1)}}{l^{(k-1)}}\right]. \tag{16}$$

Eq (16) gives the second derivative terms for all the sections, and the cubic spline curve is obtained by substituting the Eq (16) solution into Eq (14). But before that step, the boundary conditions are required for the endpoints $S''(p^{(0)})^{(0)}$ and $S''(p^{(n)})^{(n-1)}$. In the case of the mask, the reference line is closed loop; thus, a periodic condition can be applied:

$$S''(p^{(0)})^{(0)} = S''(p^{(n)})^{(n-1)}, S'(p^{(0)})^{(0)} = S'(p^{(n)})^{(n-1)}, S(p^{(0)})^{(0)} = S(p^{(n)})^{(n-1)}. \tag{17}$$

A three-dimensional cubic spline can be obtained by combining three independent cubic spline functions:

$$P(p)^{(k)} = [S_x(p)^{(k)}, S_y(p)^{(k)}, S_z(p)^{(k)}]^T, \tag{18}$$

where $P(p)^{(k)}$ is three-dimensional curve, and $S_x(p)^{(k)}, S_y(p)^{(k)}, S_z(p)^{(k)}$ are cubic spline functions to represent a basis of $P(p)^{(k)}$.

## 3. Face-mask contact model

The FE contact analysis generally includes two time-consuming procedures: finding contact areas and imposing contact constraints [39]. In this study, to achieve efficient and effective processing of the mathematical problems, five assumptions related to the face-mask contact are considered. First, the face-mask contact is considered a two-body contact problem. Second, the contact area does not change over time, and the mask exerts a constant contact pressure on the face. Third, the face-mask contact is a static problem, which is a time-independent problem; and situations of detachment, sliding, or friction of the mask over the face are not considered. Fourth, we mainly focused on the principle of contact pressure generated only along with the mask's pushing (Z-axis) direction, caused by the mask's elastics. Lastly, the mask has stiffer elasticity than human skin. Thus, the face is flexible, but the mask is assumed as rigid. These assumptions enable time-efficient computation of the face-mask contract analysis that is sufficiently effective for product design.

### 3.1. Model reduction

This study proposes a static condensation technique associated with a method of selection of master nodes for reduction of the computational cost. Static condensation [40] is a well-

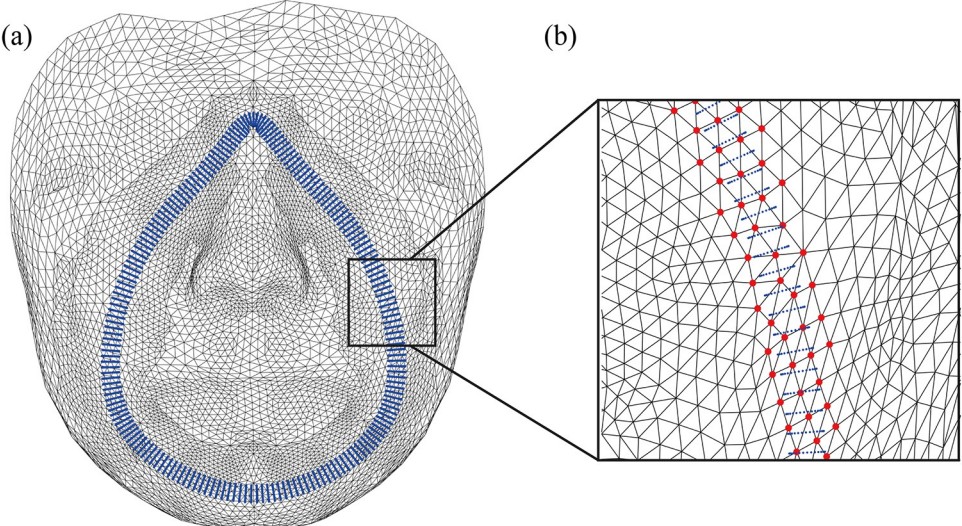

**Fig 4.** (a) Mask's nodes (blue) and (b) mater nodes of the face (red), which consist of contact elements.

known matrix technique that reduces a matrix's size by considering a few master nodes. The nodes of the face that contact the mask (interface nodes) are considered master nodes, and the others are defined as slave nodes. For the first step, the mask's nodes (blue points in Fig 4A) are placed near the face. Second, the master nodes of the face (red dots in Fig 4B) are identified as the nearest element to the mask's nodes. Third, the proposed procedure of the master node selection is applied for various design cases of the mask. Fig 5A illustrates the variation of the mask's design created by adjusting the reference points of the mask along the X-axis direction. Finally, the complete set of master nodes (red dots in Fig 5B) concerning various mask designs are selected, then the remaining nodes (black points) are defined as the slave nodes.

Using the identified master and slave nodes, the FE model of the face explained in Eq (9) is decomposed as

$$\mathbf{K}_f = \begin{bmatrix} \mathbf{K}_s & \mathbf{K}_c \\ \mathbf{K}_c^T & \mathbf{K}_m \end{bmatrix}, \quad \bar{\mathbf{u}}_f = \begin{bmatrix} \bar{\mathbf{u}}_s \\ \bar{\mathbf{u}}_m \end{bmatrix}, \quad \mathbf{f}_f = \begin{bmatrix} \mathbf{f}_s \\ \mathbf{f}_m \end{bmatrix}, \tag{19}$$

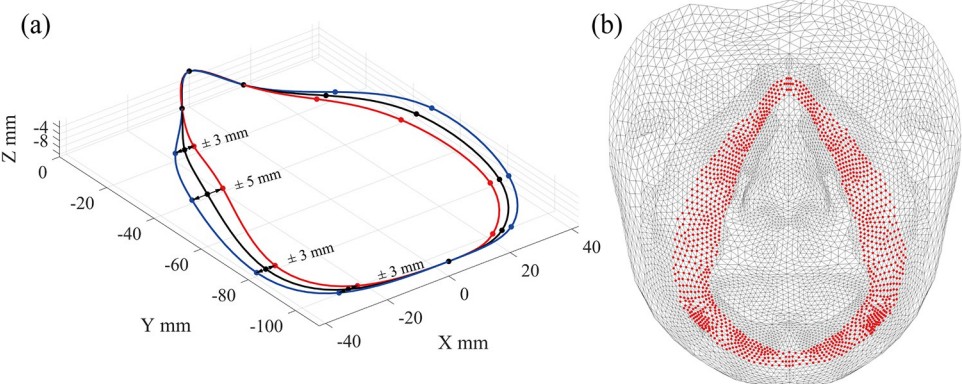

**Fig 5.** (a) Variation of the mask design and (b) the complete set of the master nodes that cover the variation of the mask design.

where the subscripts $m$ and $s$ denote the master and slave term, respectively, and the subscript $c$ presents a coupling node in the off-diagonal term. From the first row of Eq (19), the displacement of the slave nodes can be defined as that of the master displacement vector only:

$$\bar{\mathbf{u}}_s = \mathbf{K}_s^{-1}[\mathbf{f}_s - \mathbf{K}_c\bar{\mathbf{u}}_m]. \tag{20}$$

It is clear that the face-mask contact is only defined in the master node region; therefore, using Eq (20) with $\mathbf{f}_s = 0$ condition, the original displacement vector $\bar{\mathbf{u}}_f$ can be defined as

$$\bar{\mathbf{u}}_f = \mathbf{T}\bar{\mathbf{u}}_m = \begin{bmatrix} -\mathbf{K}_s^{-1}\mathbf{K}_c \\ \mathbf{I} \end{bmatrix} \bar{\mathbf{u}}_m, \tag{21}$$

where $\mathbf{T}$ is a transformation matrix. Finally, using the transformation matrix, we obtain the reduced equations as followed:

$$\mathbf{K}_r\bar{\mathbf{u}}_r = \mathbf{f}_r, \tag{22}$$

$$\bar{\mathbf{u}}_r = \bar{\mathbf{u}}_m, \quad \mathbf{K}_r = \mathbf{T}^T\mathbf{K}_f\mathbf{T}, \quad \mathbf{f}_r = \mathbf{T}^T\mathbf{f}_f, \tag{23}$$

where the subscript $r$ denotes a reduced system of the FE face model. Eq (22) clearly shows that the size of the stiffness matrix is reduced as same to the number of master displacement vector. Thus, the better effective computation is expected with small number of the master nodes than the case with the full nodes. In addition, the static problem under $\mathbf{f}_s = 0$ condition, Eq (22) provides the same solution to Eq (9) without loss of accuracy.

## 3.2. FE modeling of the face-mask contact

The uncoupled static equation of the face and mask can be expressed by recall Eq (9):

$$\begin{bmatrix} \mathbf{K}_r & 0 \\ 0 & \mathbf{K}_\gamma \end{bmatrix} \begin{bmatrix} \bar{\mathbf{u}}_r \\ \bar{\mathbf{u}}_\gamma \end{bmatrix} = \begin{bmatrix} \mathbf{f}_r \\ \mathbf{f}_\gamma \end{bmatrix}, \tag{24}$$

where the subscript $\gamma$ denotes the mask model. $\mathbf{K}_\gamma$, $\bar{\mathbf{u}}_\gamma$, and $\mathbf{f}_\gamma$ are the stiffness of the mask, the displacement of the mask, and the external force applied on the mask, respectively. In addition to this term, the definition of face-mask contact is built based on the following two conditions. (1) The Lagrange multiplier method [39], which is one of the popular contact simulation techniques, is used for the simulation of the contact effect between two bodies. (2) A node-to-node discretization method for reducing computation with a simple contact equation. The following term defines the face-mask contact model used:

$$\mathbf{C}_r^T\bar{\mathbf{u}}_r + \mathbf{C}_\gamma^T\bar{\mathbf{u}}_\gamma = \mathbf{d}_f, \quad \begin{bmatrix} \mathbf{K}_r\bar{\mathbf{u}}_r \\ \mathbf{K}_\gamma\bar{\mathbf{u}}_\gamma \end{bmatrix} + \begin{bmatrix} \mathbf{C}_r\Lambda \\ \mathbf{C}_\gamma\Lambda \end{bmatrix} = \begin{bmatrix} \mathbf{f}_r \\ \mathbf{f}_\gamma \end{bmatrix}, \tag{25}$$

$$\begin{aligned} \Lambda &= [\lambda^{(1)}, \lambda^{(2)}, \ldots, \lambda^{(N_c)}]^T, \\ \mathbf{C}_r &= [\mathbf{C}_r^{(1)}, \mathbf{C}_r^{(2)}, \cdots, \mathbf{C}_r^{(N_r)}], \\ \mathbf{C}_\gamma &= [\mathbf{C}_\gamma^{(1)}, \mathbf{C}_\gamma^{(2)}, \cdots, \mathbf{C}_\gamma^{(N_\gamma)}], \end{aligned} \tag{26}$$

where $\mathbf{C}_r$ and $\mathbf{C}_\gamma$ are Boolean matrices that choose face-mask contact nodes out of the list of face or mask nodes, respectively. $\mathbf{d}_f$ is the vector of initial distances between a face node and a corresponding mask node (say, contact pair). Since the mask is assumed as the undeformable body ($\bar{\mathbf{u}}_\gamma = 0$) in this study, thus $\mathbf{d}_f$ describes the deformation of the contact area of the face

(Eq (28)). *N* is the number of DOFs and $\mathbf{\Lambda}$ is the Lagrange multiplier vector presents contact force. From Eq (26), the augmented contact formulation is obtained as

$$\begin{bmatrix} \mathbf{K}_r & 0 & \mathbf{C}_r \\ 0 & \mathbf{K}_\gamma & \mathbf{C}_\gamma \\ \mathbf{C}_r^T & \mathbf{C}_\gamma^T & 0 \end{bmatrix} \begin{bmatrix} \bar{\mathbf{u}}_r \\ \bar{\mathbf{u}}_\gamma \\ \mathbf{\Lambda} \end{bmatrix} = \begin{bmatrix} \mathbf{f}_r \\ \mathbf{f}_\gamma \\ \mathbf{d}_f \end{bmatrix}, \tag{27}$$

then simplified to Eq (28) with $\bar{\mathbf{u}}_\gamma = 0$:

$$\begin{bmatrix} \mathbf{K}_r & \mathbf{C}_r \\ \mathbf{C}_r^T & 0 \end{bmatrix} \begin{bmatrix} \bar{\mathbf{u}}_r \\ \mathbf{\Lambda} \end{bmatrix} = \begin{bmatrix} \mathbf{f}_r \\ \mathbf{d}_f \end{bmatrix}, \tag{28}$$

which calculates the reduced displacement of the face $\bar{\mathbf{u}}_r$ by using the initial distance $\mathbf{d}_f$ and the external force $\mathbf{f}_r$.

To quantify the mask's fit, we calculate the contact pressure per node (nodal pressure).

$$f_p^{(j)} = \lambda^{(j)} / A^{(j)} = \lambda^{(j)} / \sum_{s=1}^{N_s} \frac{a^{(s)}}{3}, \tag{29}$$

where the superscript *j* denotes the *j*th contact node. $N_s$ is the number of surfaces that share a selected node. $f_p^{(j)}$ and $A^{(j)}$ are the nodal reaction force vector and the sum of areas $a^{(s)}$. Several cases for nodal pressure calculation are illustrated in Fig 6.

## 4. Numerical studies

In this section, the proposed design method was tested by considering two-step numerical problems. First, to demonstrate the adequacy of the face-mask contact effect and to exhibit characteristics of the proposed method, the structure of reference points for an initial mask design was considered. Second, to investigate the computational efficiency and applicability of the proposed method, the optimal design of the mask was sought by diversifying the designs and by analyzing the contact pressure. One of the authors participated in the present study to show expected outcomes through the proposed methods by providing a 3D scan image of the face and self-conducting usability testing. Ethical approval was not sought for the present study. A FE face model composed of 5,303 nodes and 104,963 elements was as in Section 2.1. The stiffness matrix for the face FE model $\mathbf{K}_f$ consisted of 31,818 DOFs and was prepared as a sparse matrix to minimize the data size. The algorithm was coded using MATLAB (Math-Works, Inc., Natick, MA, USA) and executed on a personal computer (Intel i7-8700K CPU, 64 GB RAM).

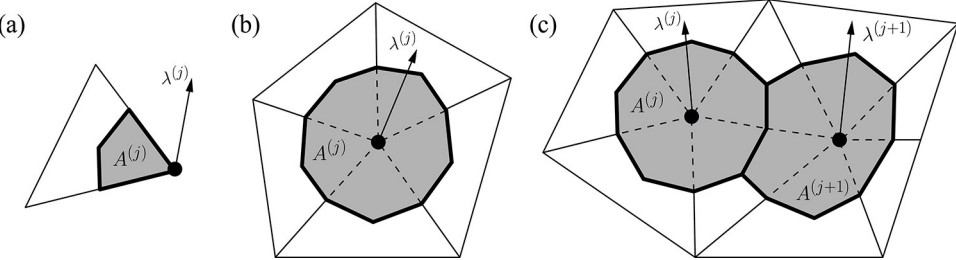

**Fig 6. Various cases of pressure calculations.** (a) One element case. (b) Five elements case. (c) Two nodes case.

## 4.1. Study on the structure of reference design points

A study on the determination of the minimum necessary number of reference design points that can be of help to the best practice of the face-mask contact simulation was conducted. Firstly, eight reference points (Fig 7A) obtained by measuring a commercial medical mask were found. The necessary nodes of a mask (Fig 7B) were initially generated along the cubic spline curve created considering the structure and number of the mask's reference points (Section 2.2). By applying the face-mask contact model presented in Section 3, the face-mask contact pressure was visually expressed, as shown in Fig 7C. The calculated contact pressure (e.g., average: 0.0092 MPa, maximum: 0.0319 MPa, as illustrated in Fig 7C) was distributed along the contact area of the mask under the condition of pushing the mask (e.g., 3 mm) toward the face. However, some areas that show zero contact pressure (0 Mpa) could occur due to the poor fit of the mask to the face (Fig 7C). This result may indicate that the spline curve was not properly drawn because the locations of eight reference points were not optimized or the number of reference points was not sufficient. In a real mask-wearing situation, an improper design

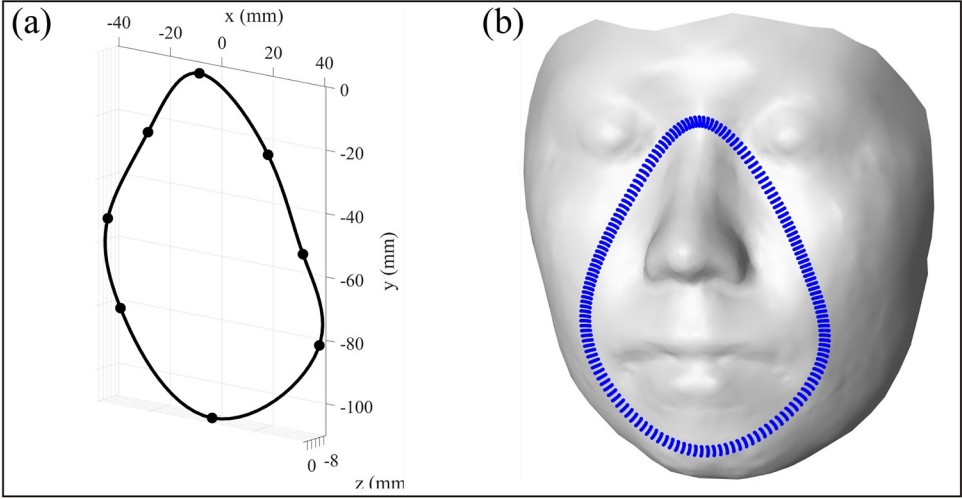

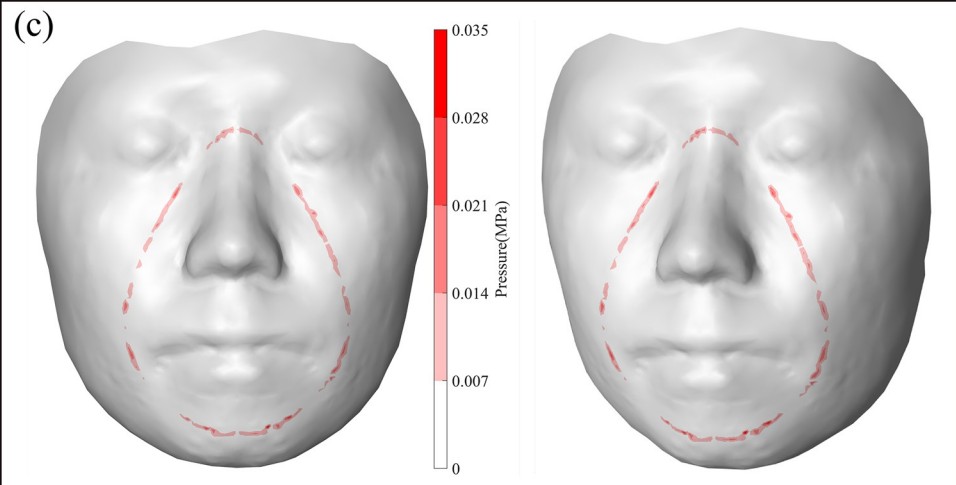

**Fig 7.** (a) Initial design of the mask generated by the cubic spline function computed using eight reference design points, (b) mask nodes along with the initial design, and (c) contact pressure result.

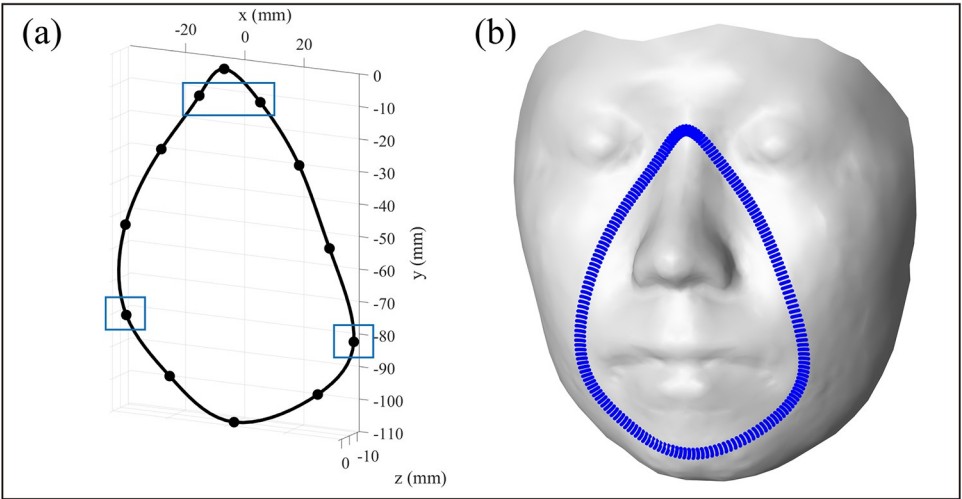

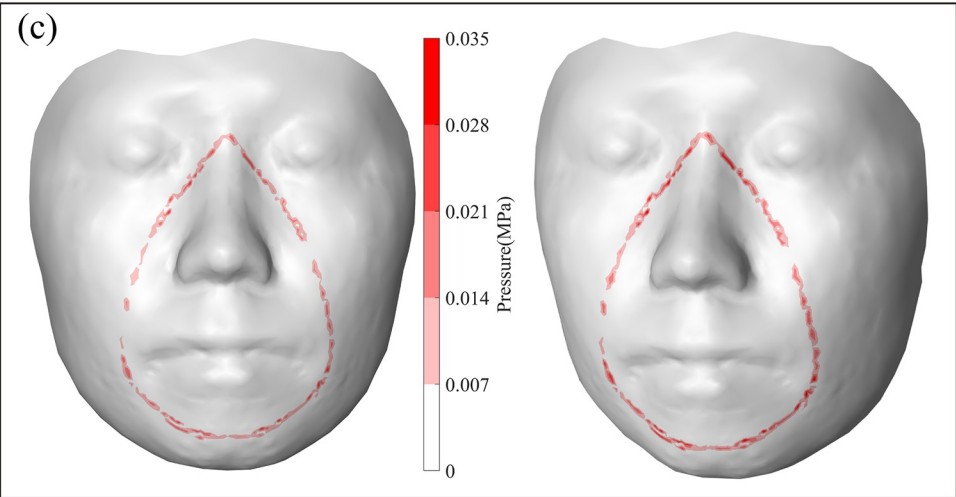

**Fig 8.** (a) A modified design of the mask generated by the cubic spline function that used 12 reference design points (blue boxes: added points), (b) mask nodes along with the initial design, and (c) contact pressure result.

can cause discomfort or pain by increasing the mask's contact pressure on the face and can allow oxygen leakage through uncontacted areas.

The poor fit of the mask was solved by increasing the number of reference points and relocating its Z-axis positions. The modified reference line was generated by adding four more reference points at the side of the nasal root and side of the lip (Fig 8A and 8B), where the face-mask contact was not properly made in the previous case. The re-calculated contact pressure (Fig 8C) showed the average and maximum values as 0.0101 MPa and 0.0282 MPa, respectively (note that it was 0.0092 MPa and 0.0319 MPa, respectively, in the eight reference points case). By increasing the curvature of the line, the result also showed a more equally distributed pressure with fewer uncontacted areas compared to the previous outcome.

### 4.2. Study on design optimization based on face-mask contact model

Then, a parametric study was performed to determine the optimal form of the mask for a particular participant's face. The mask's fit to the designated face was varied by adjusting the locations of the mask's reference points, and then contact pressure was estimated to determine the

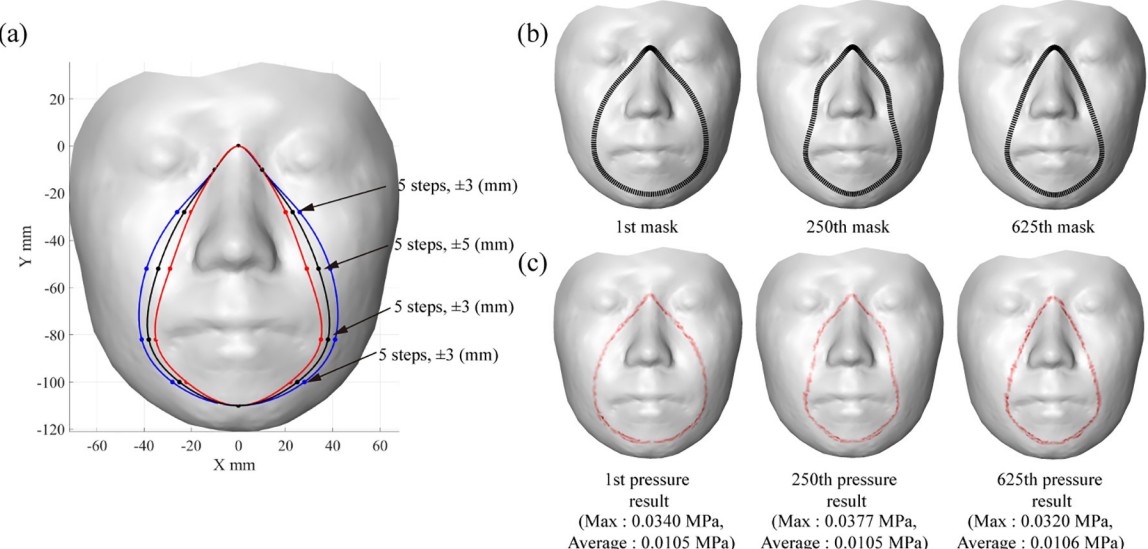

**Fig 9.** (a) Variation of mask design, (b) examples of mask nodes, and (c) results of the contact pressure (illustrated).

mask design that showed the best fit. Each reference point was adjusted based on a predefined number of steps (e.g., five steps by adjusting a point two-step toward the left and right directions from an initial location) and moving magnitudes (e.g., 3 mm) along the X-axis directions as shown in Fig 9A. Only reference points in the cheek area were adjusted in this study, and a total of 625 design candidates were tested; three are illustrated (Fig 9B and 9C). A reduced stiffness matrix composed of 1,045 master DOFs (3.3% of the total DOFs) and 30,125 slave DOFs was used to achieve computational efficiency. For the design of all 625 candidates, the time to calculate contact pressure calculation increased with the size of the stiffness matrix; it took about 28 s (0.0448 s/design) when the reduced stiffness matrix was used, but 361 s (0.5776 s/design; 12.9 times longer) when the original stiffness matrix design candidates. In addition, the average contact pressure was found to be 0.0097 ~ 0.0116 MPa throughout the designs.

A preliminary usability evaluation was included in this study to investigate the applicability and practicality of the proposed method. The selected best and worst mask designs for the participant's face were prototyped, then tested with the participant to evaluate the usability in terms of the fit and contact pressure. First, the design appropriateness of each design candidate was examined by analyzing the magnitude and distribution of the estimated contact pressure. This study assumed that a mask design showing a lower average value and lower standard deviation in the contact pressure is the most appropriate. Of the 625 design candidates, the best and the worst were then 3D-printed to test the conclusions of this study (Fig 10). Eight force-sensitive resistor (FSR) sensors (Seed Technology, Inc., South Korea) were attached at facial locations (nasal root, nasal side, cheek, chin side, and chin) to measure the magnitude and distribution of the contact pressure. By wearing a fabric face mask over the 3D-printed designs, those were naturally fitted to the face. The measured contact pressure of the best design averaged 0.0093 MPa (standard deviation: 0.008), whereas that of the worst design was 0.0150 Mpa (standard deviation: 0.009) (Fig 10C).

## 5. Discussion

This study introduced a novel face-mask contact model based on finite element analysis to design a facial mask. Rather than applying the conventional FE analysis approach that prepares

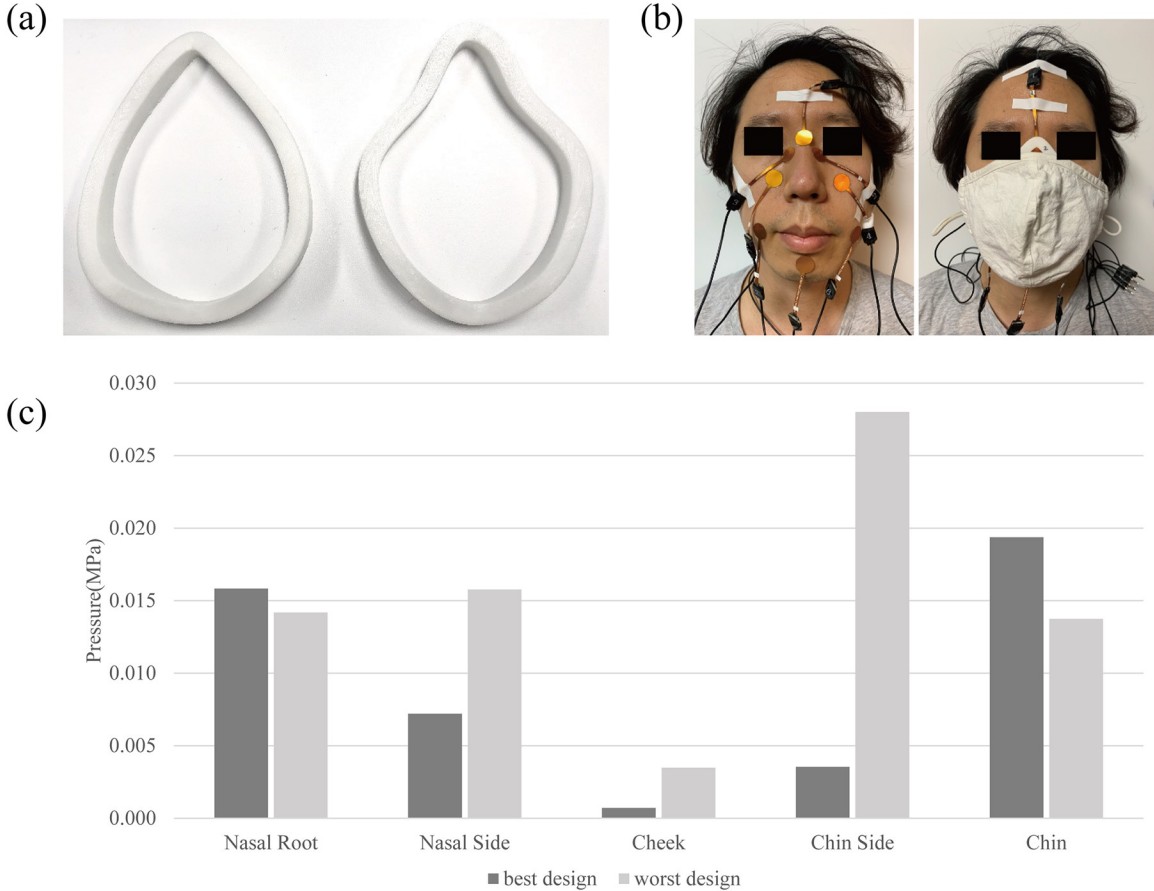

**Fig 10. Contact pressure experiment of prototyped mask design and its result.** (a) Best design (Left) and worst design (Right). (b) Eight force-sensitive resistor sensor's location. (c) Contact pressure result of the eight locations.

the geometric shape models of the body part and a contact object as realistic as possible, this study used simplified models of the face and mask in order to find the best design quickly in product design practices. The contact pressure resulting from virtually wearing a mask over the 3D scanned face was used as the evaluation metric of fit in this study, while the deformation of the contact objects is used as a fit estimator of the FE analysis in general. The contact pressure can indicate the level of discomfort, air leakage, and usability by comparing it with the actual contact pressure measured by FSR sensors. This study made three technical considerations to achieve time-efficiency of both contact pressure analysis and design customization. First, the FE model of the face was defined as a single-layered shell; thus, the face models can be quickly prepared from the 3D scanning of the face without conducting a complex process that conventional FE modeling requires. The TMR method was applied to construct a well-distributed structure of the nodes efficiently and to provide consistency in FE analyses for different faces. Second, the reduced stiffness matrix was used to decrease the computational cost effectively. Third, the adjustable model of the mask was proposed to automatically search for the optimal design through FE analysis by adjusting the number of reference points and their locations.

To effectively describe the face-mask interaction, the human face is modeled using a single-layer finite shell element, and the mask was approximated using a rigid line. Further reduction of computational cost is available by using static condensation, which is a well-known

technique to reduce FE models [40]. In this manner, the face-mask contact can be simulated around a small contact area. As a result, only 3.3% of total DOFs in the numerical model (Chapter 4) were required to compute the contact pressure. The advantage of the proposed approach increases when the representative design should be computed for certain target user groups of over thousands of people; in this case, the approach of using a specific FE face model that considers an individual's anatomic characteristics (e.g., the thickness of soft tissue, the shape of a bone, material properties) might be inefficient. It should also be noticed that considering human biological uncertainties and various nonlinearities is difficult, although specific FE faces modeling is employed. On the other hand, the proposed algorithm can provide numerous reasonable results in a short computation time.

The face-mask contact model proposed in this study was found that it is effective and practical in terms of the time and cost of designing and evaluating facial masks. A total of 625 mask design candidates were systematically assessed using the virtual fit simulation that uses the proposed face-mask contact model. Then, the fit simulation automatically seeks the optimal form of a mask for an individual face in no time (28 seconds for 625 design candidates) by calculating the contact pressure. Also, compared to the previous face-mask contact studies [19], the contact pressure predicted in the fit simulation showed similar order to that obtained through the pilot usability study in terms of the average (mean difference: 0.0005 MPa) and variation (range: 0.0087 to 0.0804 Mpa). It could explain that the face-mask contact model might be appropriately designed based on the aforementioned technical considerations. Moreover, the present study would be effective in terms of the cost in the product design process as the product design generally requires a cost-consuming physical prototyping phase to test the usability of designs. On the other hand, the proposed method of virtually testing and searching for the most appropriate design candidates in terms of the fit and contact pressure could save the time and cost of iterative prototyping and testing in the product design process [41].

This study concentrated on the efficiency of the face-mask contact analysis by simplifying the FE model with appropriate assumptions, but further studies on the following issues are needed to enhance the method. First, only one face was tested through the pilot study to validate the proposed FE model; but the similarity between the experimental and simulated results with various faces should be further studied to advance the FE model more accurately and reliably. Second, to design a mask product for a particular target population, the proposed face-mask contact model needs to quickly provide a proper design guideline based on engineering information in a mask design process. In further work, the probability functions for design parameters such as the face, material properties, and device shape will be considered using the sampling method and metamodeling in the computation step. Thus, a stochastic result regarding the contact pressure can be found, and the most proper mask shape for the target user population can be systematically obtained.

The proposed algorithm provides various research potentials as follows. First, in the proposed method, the FE mesh quality and the reliability of the material properties are the major factors of accuracy. The proper mesh quality can be achieved by TMR with the 3D scan images. However, in this work, the same material properties are used at all locations in the face, and this uniformity may cause an inaccurate calculation of contact pressure. Dividing a face model into several areas that have different material properties (such as bone-dominant and tissue-dominant) is a remedy to handle this issue. Using the proposed method, virtual fitting results over thousands of people can be easily obtained in a short time, and then the best virtual prototype can be identified by comparing the estimated contact pressures. Therefore, the probability-based design software (or framework) to find an optimal shape for target user databases could be developed based on the proposed algorithm. In addition, we only focused on the face-mask interaction in this work, but the proposed method could be extended to various

wearable devices such as earphones, AR glasses, smart watches, etc. However, the contact assumptions considered in this work are valid only if wearable devices are much stiffer than the skin and undergo small deformation. Other contact approaches to cover two flexible bodies are required if the deformation of the wearable devices becomes significant.

## 6. Conclusion

The main theme of this study was to use numerous 3D human images to check the fit of design candidates quickly in a product-design process. The combination of data parameterization by template model registration, contact pressure estimation by FE analysis method, and computation simplification by static condensation techniques was proposed and applied to achieve reliable and efficient calculation of the face-mask contact pressure. The proposed method may contribute to the ergonomic design of a facial wearable product such as various types of masks (e.g., medical mask, dust-proof mask, gas mask, protective mask, sports mask), and goggles (e.g., protective goggles, sports goggles, virtual reality headset) that require a good fit and comfort to provide health, safety, and satisfaction to users. Also, the proposed method could contribute to developing an engineering design tool (e.g., CAD) or automated manufacturing, which can effectively support practitioners in a product design and development process.

## Author Contributions

**Conceptualization:** Jin-Gyun Kim, Wonsup Lee.

**Data curation:** Yun-Jae Kwon, Wonsup Lee.

**Formal analysis:** Yun-Jae Kwon, Wonsup Lee.

**Funding acquisition:** Jin-Gyun Kim, Wonsup Lee.

**Methodology:** Yun-Jae Kwon, Jin-Gyun Kim.

**Project administration:** Wonsup Lee.

**Software:** Yun-Jae Kwon.

**Supervision:** Jin-Gyun Kim, Wonsup Lee.

**Validation:** Wonsup Lee.

**Visualization:** Yun-Jae Kwon, Wonsup Lee.

**Writing – original draft:** Yun-Jae Kwon.

**Writing – review & editing:** Yun-Jae Kwon, Jin-Gyun Kim, Wonsup Lee.

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
