## [Decision Letter · Decision Letter 0]

4 Apr 2022

PONE-D-22-05499A framework for effective face-mask contact modeling based on finite element analysis for custom design of a facial maskPLOS ONE

Dear Dr. Lee,

Thank you for submitting your manuscript to PLOS ONE. After careful consideration, we feel that it has merit but does not fully meet PLOS ONE’s publication criteria as it currently stands. Therefore, we invite you to submit a revised version of the manuscript that addresses the points raised during the review process.

We look forward to receiving your revised manuscript.

Kind regards,

Marko Čanađija

Academic Editor

PLOS ONE

Journal Requirements:

[This research was jointly supported by the National Research Foundation of Korea (NRF) grants funded by Korea Government (2020R1F1A1050076 and 2021R1A2C4087079).]

 [Acknowledgement

- This research was jointly supported by the National Research Foundation of Korea (NRF) grants funded by Korea Government (2020R1F1A1050076 and 2021R1A2C4087079).

Initials and grant numbers

- WL: 2020R1F1A1050076

- JGK: 2021R1A2C4087079

Full name of each funder

- the National Research Foundation of Korea (NRF)

(URL: https://www.nrf.re.kr/eng/index)

Did the sponsors or funders play any role in the study design, data collection and analysis, decision to publish, or preparation of the manuscript?

- NO. The funders had no role in study design, data collection and analysis, decision to publish, or preparation of the manuscript.]

Additional Editor Comments:

Some reviewers have proposed several papers for inclusion in the bibliography. Please include those papers only if they are related to the topic of the manuscript.

Reviewers' comments:

Reviewer's Responses to Questions

**Comments to the Author**

1. Is the manuscript technically sound, and do the data support the conclusions?

Reviewer #1: Yes

Reviewer #2: Yes

2. Has the statistical analysis been performed appropriately and rigorously? 

Reviewer #1: I Don't Know

Reviewer #2: I Don't Know

3. Have the authors made all data underlying the findings in their manuscript fully available?

Reviewer #1: No

Reviewer #2: Yes

4. Is the manuscript presented in an intelligible fashion and written in standard English?

Reviewer #1: Yes

Reviewer #2: Yes

5. Review Comments to the Author

Reviewer #1: This manuscript studied the face-mask contact behaviour using finite element analysis. As claimed by the authors, the motivation of this study is to provide guidelines to optimise the design of face masks. In my opinion, the paper can be accepted, provided the following concerns are addressed.

The background and motivation need enhancement.

This paper's efficacy heavily depends on several assumptions. The simplified model is subsequently adopted, which reduces computational effort. I am curious about how the authors verify the result. I will suggest putting more emphasis on the verification part.

Will the implanted electronics affect the contact performance?

Other scholars have also looked into mask-airway contact behaviours. For example, Liao H, et al. Sealing mechanism study of laryngeal mask airways via 3D modelling and finite element analysis[J]. Scientific Reports. What are the difference, mainly the benefits, of using the approaches developed in this paper?

The paper needs thorough proofreading to remove the grammatical error.

Reviewer #2: The authors performed “FE model of the face” but I couldn’t FE model outputs (stress and deformation) in manuscript. They should be showed the outputs.

Today, computer aided finite element analysis (FEA) was used to solve processes simulations of COVID-19 and other infections and biomechanics [1-5]. The authors should be cited the following references to enrich of paper.

References

1. ERDEM M, GOK K, GOKCE B, GOK A (2017) NUMERICAL ANALYSIS OF TEMPERATURE, SCREWING MOMENT AND THRUST FORCE USING FINITE ELEMENT METHOD IN BONE SCREWING PROCESS. Journal of Mechanics in Medicine and Biology 17 (01):1750016. doi:10.1142/s0219519417500166

2. Pirhan Y, Gök K, Gök A (2020) Comparison of two different bowel anastomosis types using finite volume method. Computer Methods in Biomechanics and Biomedical Engineering 23 (8):323-331. doi:10.1080/10255842.2020.1722809

3. Gök K, Selçuk AB, Gök A (2021) Computer-Aided Simulation Using Finite Element Analysis of Protect Against to Coronavirus (COVID-19) of Custom-Made New Mask Design. Transactions of the Indian Institute of Metals. doi:10.1007/s12666-021-02227-4

4. Gok K Investigation of the use of silicone pads to reduce the effects on the human face of classical face masks used to prevent from COVID-19 and other infections. Proceedings of the Institution of Mechanical Engineers, Part E: Journal of Process Mechanical Engineering 0 (0):09544089211019581. doi:10.1177/09544089211019581

5. Gok K, Erdem M, Kisioglu Y, Gok A, Tumsek M (2021) Development of bone chip-vacuum system in orthopedic drilling process. Journal of the Brazilian Society of Mechanical Sciences and Engineering 43 (4):224. doi:10.1007/s40430-021-02959-w

6. PLOS authors have the option to publish the peer review history of their article (what does this mean?). If published, this will include your full peer review and any attached files.

Reviewer #1: No

Reviewer #2: No

---

## [Author Response · Author response to Decision Letter 0]

19 May 2022

The responses to the comments are also provided as an enclosed file (revision note.docx).

Editor comment

⇒ We have checked and corrected the style of the manuscript.

⇒ As we are preparing the technology for a commercial purpose, we would not want to share our system directly. But, mathematic algorithms for the core concept of our technology have been explained in the manuscript clearly and anyone in this field might easily understand and apply the concept in their studies.

3. Funding information should not appear in the Acknowledgments section or other areas of your manuscript. We will only publish funding information present in the Funding Statement section of the online submission form. Please remove any funding-related text from the manuscript and let us know how you would like to update your Funding Statement. Please include your amended statements within your cover letter; we will change the online submission form on your behalf.

⇒ The acknowledgement section is removed in the updated manuscript. And there’s no updates for the current Funding Statement.

⇒ We have published our data resulting from the FE simulation shown in our study, and here is the DOI number: http://dx.doi.org/10.13140/RG.2.2.21674.49607.

Reviewer 1

1. The background and motivation need enhancement.

⇒ We have enhanced the background by making detailed descriptions in the introduction. More relevant studies have been added in the 3rd paragraph of the introduction. The 4th paragraph of the introduction has been newly added, and the 5th paragraph of the introduction has been updated to emphasize our motivation for the study.

2. This paper's efficacy heavily depends on several assumptions. The simplified model is subsequently adopted, which reduces computational effort. I am curious about how the authors verify the result. I will suggest putting more emphasis on the verification part.

⇒ In our study, the verification was performed through an experiment of measuring the contact pressure using the FSR sensors, which is described in the Numerical Studies section. The values were compared using their mathematical order. Since the measured data was generated from the different masks and face shapes, we found that the contact pressure level and distribution of the algorithm are well-matched with the reference data. For example, the results of the contact pressure analysis in other papers indicate that the contact pressure shows a value between 0.001 MPa to 0.1 MPa in the cases of face-mask contact analysis found in the previous studies. And our results showed similar order to that obtained by the experiment in terms of the average (mean difference: 0.0005 MPa) and variation (range: 0.0087 to 0.0804 MPa), as discussed in the discussion section. However, as we tested for the one face through the pilot study, the similarity between the experimental and simulated results with various faces should be further studied to define the FE model more precisely. We newly added the 4th paragraph in the discussion to explain limitations and future studies.

3. Will the implanted electronics affect the contact performance?

⇒ Our proposed method simply calculates the contact pressure only between two surfaces (body surface and product surface) data without considering the mass of the human body or product. Therefore, the effect of the implanted electronics contacting inside of the human soft tissue is unavailable to estimate through our method.

4. Other scholars have also looked into mask-airway contact behaviours. For example, Liao H, et al. Sealing mechanism study of laryngeal mask airways via 3D modelling and finite element analysis. Scientific Reports. What are the difference, mainly the benefits, of using the approaches developed in this paper?

⇒ The goal of the presented method is to solve the face-mask contact problem efficiently using the simplified FE model and FE analysis procedure. Our method can compute the contact pressure of hundreds of cases in a short time. Thus, product designers can understand the contact pressure characteristics easily by changing design parameters. The quantified information resulting from our method can guide designers to obtain the best shape of a designated product quickly.

5. The paper needs thorough proofreading to remove the grammatical error.

⇒ We found the grammatical errors and the manuscript was proofread.

Reviewer 2 comment

1. The authors performed “FE model of the face” but I couldn’t see FE model outputs (stress and deformation) in manuscript. They should be showed the outputs.

⇒ The displacement and stress commonly used as estimators in FE analysis were considered in our mathematic process (equation 1); but, we used the contact pressure as an indicator of discomfort, air leakage, and usability. As the FE model can be verified by using the contact pressure as it can also be measured by sensors in a real situation. Also, this study didn’t calculate the deformation to reduce the computational load in the proposed procedure. Therefore, displacement and stress were not used in our framework. The following figure shows stress, displacement, and contact pressure showing a similar trend. We have added about this issue in the 5th paragraph of the introduction and the 1st paragraph of the discussion sections.

2. Today, computer aided finite element analysis (FEA) was used to solve processes simulations of COVID-19 and other infections and biomechanics [1-5]. The authors should be cited the following references to enrich of paper.

⇒ We have cited the mentioned papers in the introduction.

---

## [Decision Letter · Decision Letter 1]

6 Jun 2022

A framework for effective face-mask contact modeling based on finite element analysis for custom design of a facial mask

PONE-D-22-05499R1

Dear Dr. Lee

We’re pleased to inform you that your manuscript has been judged scientifically suitable for publication and will be formally accepted for publication once it meets all outstanding technical requirements.

Kind regards,

Marko Čanađija

Academic Editor

PLOS ONE

Additional Editor Comments (optional):

Reviewers' comments:

Reviewer's Responses to Questions

**Comments to the Author**

1. If the authors have adequately addressed your comments raised in a previous round of review and you feel that this manuscript is now acceptable for publication, you may indicate that here to bypass the “Comments to the Author” section, enter your conflict of interest statement in the “Confidential to Editor” section, and submit your "Accept" recommendation.

Reviewer #1: All comments have been addressed

2. Is the manuscript technically sound, and do the data support the conclusions?

Reviewer #1: Yes

3. Has the statistical analysis been performed appropriately and rigorously? 

Reviewer #1: N/A

4. Have the authors made all data underlying the findings in their manuscript fully available?

Reviewer #1: Yes

5. Is the manuscript presented in an intelligible fashion and written in standard English?

Reviewer #1: Yes

6. Review Comments to the Author

Reviewer #1: The authors have honestly answered my concerns. The paper brings some novelty. I have no further comments.

7. PLOS authors have the option to publish the peer review history of their article (what does this mean?). If published, this will include your full peer review and any attached files.

Reviewer #1: No

---

## [Editor Report · Acceptance letter]

28 Jun 2022

PONE-D-22-05499R1 

A framework for effective face-mask contact modeling based on finite element analysis for custom design of a facial mask 

Dear Dr. Lee:

I'm pleased to inform you that your manuscript has been deemed suitable for publication in PLOS ONE. Congratulations! Your manuscript is now with our production department. 

Kind regards, 

on behalf of

Dr. Marko Čanađija 

Academic Editor

PLOS ONE